High genetic diversity and mixing of coastal horseshoe crabs (Tachypleus gigas) across major habitats in Sundaland, Indonesia

Aini Naila Khuril 1
Wardiatno Yusli 2 3
Effendi Hefni 2 3
Mashar Ali 2
Madduppa Hawis hawis@apps.ipb.ac.id 4 5
1 Study Program of Aquatic Resources, Department of Aquatic Resources Management, Faculty of Fisheries and Marine Sciences, Graduate School, Institut Pertanian Bogor , Bogor , Jawa Barat , Indonesia
2 Department of Aquatic Resources Management, Faculty of Fisheries and Marine Sciences, Institut Pertanian Bogor , Bogor , Jawa Barat , Indonesia
3 Environmental Research Center, Institut Pertanian Bogor , Bogor , Indonesia
4 Department of Marine Science and Technology, Faculty of Fisheries and Marine Sciences, Institut Pertanian Bogor , Bogor , Jawa Barat , Indonesia
5 Oceanogen Environmental Biotechnology Laboklinikum , Bogor , Indonesia
Reimer James
Electronic publication date: 2021 Aug 2
Publication date: 2021
Volume: 9
Electronic Location ID: e11739
Received 2020 Nov 14; Accepted 2021 Jun 17
Copyright: ©2021 Aini et al.
Copyright year: 2021
Copyright holder: Aini et al.
License: This is an open access article distributed under the terms of the Creative Commons Attribution License, which permits unrestricted use, distribution, reproduction and adaptation in any medium and for any purpose provided that it is properly attributed. For attribution, the original author(s), title, publication source (PeerJ) and either DOI or URL of the article must be cited.
License URL: https://creativecommons.org/licenses/by/4.0/

Keywords: Population genetics, Zoogeography, Biogeography, Molecular biology, Endangered species, Protected species, Coral triangle, Population structure, Meta-population

Funding: Research Center for Oceanography Indonesian Institute of Sciences (PPO-LIPI) through Demand-Driven Research Fund B-5063/IPK.2/KS.02/III/2019 SEAMEO BIOTROP DIPA 2020 No. 047.35/PSRP/SC/SPK-PNLT/III/2020 PMDSU 6623/IT3.L1/PN/2019 The Ministry of Research, Technology and Higher Education 261/SP2H/LT/DRPM/20190 This work was supported by the Research Center for Oceanography, Indonesian Institute of Sciences (PPO-LIPI) through Demand-Driven Research Fund Contract No. B-5063/IPK.2/KS.02/III/2019, SEAMEO BIOTROP DIPA 2020 No. 047.35/PSRP/SC/SPK-PNLT/III/2020, PMDSU grant Contract No. 6623/IT3.L1/PN/2019, and the Ministry of Research, Technology and Higher Education No. 261/SP2H/LT/DRPM/20190). The funders had no role in study design, data collection and analysis, decision to publish, or preparation of the manuscript.

==============================
Species with limited dispersal abilities are often composed of highly genetically structured populations across small geographic ranges. This study aimed to investigate the haplotype diversity and genetic connectivity of the coastal horseshoe crab (Tachypleus gigas) in Indonesia. To achieve this, we collected a total of 91 samples from six main T. gigas habitats: Bintan, Balikpapan, Demak, Madura, Subang, and Ujung Kulon. The samples were amplified using primers for mitochondrial (mt) AT-rich region DNA sequences. The results showed 34 haplotypes, including seven shared and 22 unique haplotypes, across all localities. The pairwise genetic differentiation (FST) values were low (0 to 0.13) and not significantly different (p > 0.05), except among samples from Ujung Kulon-Madura and Kulon-Subang (p < 0.05). Additionally, the 34 analysis of molecular variance (AMOVA) showed the most variation within populations (95.23%) compared to less among populations (4.77%). The haplotype network showed evidence of shared haplotypes between populations. Tajima’s D and Fu’s FS test values indicated a population expansion. Our results showed a low level of differentiation, suggesting a single stock and high connectivity. Therefore, a regionally-based conservation strategy is recommended for the coastal horseshoe crab in Indonesia.

Introduction

High rates of gene flow are common in marine organisms that are spread across large geographic ranges (Palumbi, 1994; Crandall et al., 2019). Several marine organisms also exhibit low levels of genetic differentiation across large geographic scales (Avise, 2000). Population structures are affected by genetic drift, strong post-settlement selection (Hedgecock, 1986), and spatial-landscape patterns (Johnson & Black, 1998; Watts & Johnson, 2004). Species with limited dispersal abilities are often composed of highly genetically structured populations with small geographic ranges (Collin, 2001). This creates opportunities to compare the depths and positions of intraspecific genetic differentiation when using location as an extrinsic factor (Bernardi & Talley, 2000).

Horseshoe crabs, an interesting group of marine organisms considered “living fossils” (Eldredge & Stanley, 1984), have been extant for almost 500 million years. There are four extant species of horseshoe crabs: the American horseshoe crab (Limulus polyphemus) found along the eastern coast of North America from Maine to Mexico (Walls, Berkson & Smith, 2002; Rutecki, Carmichael & Valiela, 2004), and three Asian horseshoe crabs species (the mangrove horseshoe crab (Carcinoscorpius rotundicauda), the coastal horseshoe crab (Tachypleus gigas), and the tri-spined horseshoe crab (Tachypleus tridentatus)) (John et al., 2018; Vestbo et al., 2018) that are sporadically distributed across Southeast Asia and Japan. They are ancient marine arthropods that exhibit life-histories and habitat preferences that suggest a restricted dispersal ability (Sekiguchi, 1988). The Asian species are found in Indonesian coastal waters, dispersed around Sumatra, Java, Kalimantan, and Sulawesi (Rubiyanto, 2012; Mashar et al., 2017; Meilana et al., 2016).

Throughout their life cycle, horseshoe crabs are highly dependent on environmental conditions in coastal habitats. Most research suggests that they are declining both locally and regionally due to over-harvesting for food and biomedicine, and coastal development (Itow, 1993; Botton, 2001; Chen, Yeh & Lin, 2004) and the loss of suitable spawning grounds. T. gigas was once relatively common along the northern Java Sea. However, coastal and mangrove horseshoe crab populations have an undetermined conservation status due to insufficient data (John et al., 2021). Furthermore, most population genetic studies on horseshoe crabs have focused on the American horseshoe crab, with little attention paid to the Asian horseshoe crab (Pierce, Tan & Gaffney, 2000; King & Eackles, 2004; King et al., 2005; Yang et al., 2007; Rozihan & Ismail, 2011; King et al., 2015). Therefore, this study examined the genetic diversity, connectivity, and population structure of coastal horseshoe crabs by screening an AT-rich region of mitochondrial DNA, an established genetic marker for arthropods (Brehm et al., 2001). Our aim was to use genetic evidence to facilitate horseshoe crab conservation efforts in Indonesia.

Materials & Methods

Study area and sample collection

With the help of a local fisherman, adult and juvenile T. gigas specimens were collected from shallow waters in six locations around Indonesia: Bintan, Balikpapan, Demak, Madura, Subang, and Ujung Kulon (Fig. 1). We collected the hemolymph from a total of 91 T. gigas specimens between April 2019 and August 2020. There were eight, 14, 16, 13, 20, and 20 samples from Bintan Island (BT), Balikpapan (BP), Demak (DK), Madura (MD), Subang (SB), and Ujung Kulon (UK), respectively. The hemolymph was collected from each individual and immediately preserved in absolute ethanol. Field experiments were approved by the Research Council of the Study Program from IPB University (letter number 1426/IT3.F3.2/KP.03.03.2019).

Figure 1 Sampling locations of Tachypleus gigas; There were eight, 14, 16, 13, 20, and 20 samples from Bintan Island (BT) = 8, Balikpapan (BP) = 14, Demak (DK) = 16, Madura (MD) = 13, Subang (SB) = 20 and Ujung Kulon (UK) = 20.

Genomic DNA extraction, amplification, and DNA sequencing

Genomic DNA was isolated from each hemolymph sample following a Genomic DNA Mini Kit (Geneaid, New Taipe, Taiwan) according to the manufacturer’s instructions. A fragment of the AT-rich region was amplified using a pair of primers, Hb-12S (5′-GTCTAACCGCGGTAGCTGGCAC-3′) and Hb-trna (5′GAGCCCAATAGCTTAAATTAGCTTA-3′), designed from the mitochondrial genome of the American horseshoe crab (Lavrov, Boore & Brown, 2000). A 25-µL PCR reaction was carried out with 12.5 µL MyTaq HS Red Mix (Meridian Bioscience, OH, United States), 9 µL ddH2O, 1.25 µL forward and reverse primer, and 1 µL DNA template. The entire reaction mixture was amplified using a peqSTAR thermal cycler (Peqlab, Erlangen, Germany), following Yang et al.’s (2007) amplification steps. The mixture underwent pre-denaturation at 95 °C for 3 mins, followed by 30 cycles of denaturation at 94 °C for 30 s, annealing at 50 °C for 1 min, extension at 72 °C for 2 min, one cycle at 72 °C for 2 min, and 25 °C for 5 min. The PCR product was visualized using electrophoresis on a 1% agarose gel in TAE buffer with ethidium bromide at 100 V for 30 min. After electrophoresis, the gel was placed under UV light for band detection to determine the presence of a DNA fragment. The DNA sequencing was performed by 1st BASE DNA Sequencing Services, Selangor, Malaysia.

Data analysis

A total of 91 AT-rich region sequences were obtained, and MEGA X (Kumar et al., 2018) was used to generate multiple alignments of the edited sequences. Genetic diversity was measured using the number of haplotypes (Hn), haplotype diversity (Hd) and nucleotide diversity (π) using DNASp v6 (Rozas et al., 2017). The population structure was assessed using Wright’s fixation index (FST) and analysis of molecular variance (AMOVA). The significance level threshold (α), used to determine the pattern of differentiation between locations, was 0.05. The pairwise F-statistic (FST) was calculated as the genetic distance based on the population differences using DNASp v6 (Rozas et al., 2017). The haplotype network across populations was estimated using a median joining (MJ) network (Bandelt, Forster & Röhl, 1999) and was calculated using Network v 4.6.1.0 based on haplotype data. The haplotype composition across all study areas was illustrated in a map to show distribution and genetic connectivity patterns across the populations. Tajima’s D (1989) and Fu’s FS (1997) statistical tests were used to assess the population equilibrium using the Arlequin v.3.5 program (Excoffier & Lischer, 2010).

Results

Genetic diversity

We obtained a total of 91 AT-rich sequences of approximately 670 bp across all sampling locations including Java (UK, SB, DK, and MD), Sumatra, Bintan and Borneo (Balikpapan). In total, 43 variable nucleotide sites and 34 haplotypes were observed. The haplotypes consisted of both unique (found only in certain locations) and common haplotypes (Table 1). The genetic diversity of the coastal horseshoe crab varied across sampling sites (Table 2). The percentage of A+T composition at each location, which differed slightly, was approximately 81%.

Table 1 Variable sites found in a fragment of the AT-rich region of Tachypleus gigas in each populations.

Fourty three variable sites were found in a fragment of the AT-rich region in 91 horseshoe crabs defining 34 haplotypes (H1–H34).

	Nucleotide positions	n	
				1	1	1	2	2	2	2	2	2	3	3	3	3	3	3	3	3	3	3	4	4	4	4	4	4	4	4	4	4	4	4	5	5	5	5	5	6	6	6	6		
	2	3	8	3	7	7	6	6	6	6	7	8	1	4	4	4	5	6	7	7	7	8	0	1	1	1	3	3	4	6	7	7	9	9	0	6	6	6	8	2	4	7	8		
	5	2	3	5	4	8	0	1	6	9	4	2	3	2	3	4	9	6	2	3	4	6	1	3	4	5	0	7	6	7	2	7	1	2	2	3	6	7	9	0	7	2	5		
H1	T	T	C	C	C	T	G	A	C	A	C	T	T	C	A	A	C	T	T	A	T	A	C	T	T	T	G	A	T	T	A	A	A	C	C	T	A	A	G	C	T	G	C	7	
H2	C	C	.	.	.	C	A	.	.	.	A	.	.	.	.	.	.	.	A	.	.	.	.	.	.	.	.	G	.	C	.	.	.	T	.	.	.	.	.	A	A	.	.	1	
H3	C	C	.	.	.	C	A	.	.	.	.	.	.	.	.	.	T	.	.	.	.	.	.	.	.	.	.	.	.	.	.	.	.	.	.	.	.	.	.	.	A	.	.	15	
H4	.	.	.	.	.	.	.	.	.	.	.	.	.	.	.	.	.	.	.	.	C	.	.	.	.	.	.	.	.	.	.	.	.	T	.	.	.	.	.	.	.	.	.	1	
H5	.	C	.	.	.	.	.	.	.	.	.	.	.	.	.	.	.	.	.	.	.	.	.	.	.	.	.	.	.	.	.	.	.	.	.	.	.	.	.	.	.	.	.	12	
H6	.	C	.	.	.	.	.	.	.	.	.	.	.	.	.	.	.	.	.	.	.	.	.	.	.	.	.	.	.	.	G	.	.	.	.	.	.	.	.	.	.	.	.	2	
H7	.	C	.	.	.	.	.	.	.	.	.	.	.	.	.	.	.	.	.	.	.	.	.	.	.	.	.	.	.	C	.	.	.	.	T	.	.	.	.	.	.	.	.	3	
H8	C	C	.	.	.	C	A	.	.	.	.	.	.	.	.	.	T	.	.	.	.	.	.	.	.	.	.	.	.	.	.	.	.	T	.	.	.	.	.	.	A	.	.	5	
H9	C	C	.	.	.	C	A	.	.	.	.	.	.	.	.	.	.	.	.	.	.	.	.	.	.	.	.	.	.	.	.	.	.	.	.	.	.	.	.	.	A	.	.	6	
H10	C	C	T	.	T	C	A	G	.	.	.	C	.	T	G	T	.	.	.	.	C	.	T	.	C	C	A	.	.	.	.	.	.	.	.	.	.	.	.	.	A	.	.	1	
H11	C	C	.	.	.	C	A	.	.	.	.	.	.	.	.	.	T	C	.	.	.	.	.	.	.	.	.	.	.	C	.	.	.	.	.	.	.	.	.	.	A	.	.	1	
H12	.	C	.	.	.	.	.	.	.	.	.	.	.	.	.	.	.	.	.	.	.	.	.	.	.	.	.	.	.	.	.	.	.	.	.	.	.	G	.	.	.	.	.	1	
H13	C	C	T	.	.	C	A	.	T	.	.	C	.	.	.	T	.	.	.	.	.	G	.	.	.	.	.	.	.	C	.	.	G	.	.	.	G	.	.	.	A	.	.	1	
H14	.	C	.	.	.	.	.	.	.	.	.	.	.	.	.	.	.	.	.	G	.	.	.	.	.	.	.	.	.	.	.	.	.	.	.	.	.	.	.	.	.	.	.	2	
H15	C	C	.	.	T	C	A	.	.	.	.	.	.	.	.	.	T	.	.	.	.	.	.	.	.	.	.	.	.	.	.	.	.	T	.	.	.	.	.	.	A	.	.	1	
H16	.	.	.	.	.	.	A	.	.	.	.	.	.	.	.	.	.	.	.	.	.	.	.	C	.	.	.	.	.	.	.	.	.	.	.	.	.	.	.	.	.	.	.	1	
H17	C	C	.	.	.	C	A	.	.	.	.	.	.	.	.	.	T	.	.	.	.	.	.	.	.	.	.	.	C	.	.	.	.	.	.	.	.	.	.	.	A	.	.	2	
H18	.	C	.	.	.	.	.	.	.	.	.	.	C	.	.	.	.	.	.	.	.	.	.	.	.	.	.	.	.	.	.	.	.	.	.	.	.	.	.	.	.	.	.	9	
H19	.	C	.	.	.	.	.	.	.	.	.	.	C	.	.	.	.	.	.	.	.	.	.	.	.	.	.	.	C	C	.	.	.	.	.	.	.	.	.	.	.	.	.	1	
H20	.	C	.	.	.	.	.	.	.	.	.	.	C	.	.	.	.	.	.	.	.	.	.	.	.	.	.	.	.	C	.	.	.	.	.	.	.	.	.	.	.	.	.	1	
H21	C	C	T	.	T	C	A	.	.	.	.	C	.	.	.	T	.	.	.	.	.	G	.	.	.	.	.	.	.	.	.	.	.	.	.	.	.	.	.	.	A	.	.	1	
H22	C	C	.	.	.	C	A	.	.	.	.	.	.	.	.	.	T	.	.	.	.	.	.	.	.	.	.	.	.	.	C	.	.	.	.	.	.	.	.	.	A	.	.	1	
H23	C	C	.	.	.	C	A	.	.	.	.	.	.	.	.	.	.	.	.	.	.	.	.	.	.	.	.	.	.	.	.	.	.	.	.	.	.	.	A	.	A	.	.	3	
H24	C	C	.	.	.	C	A	.	.	.	.	.	.	.	.	.	.	.	.	.	.	.	.	.	.	.	.	.	.	.	.	.	.	.	.	.	.	.	.	.	A	A	.	1	
H25	C	C	.	.	.	C	A	.	.	.	.	.	.	.	.	.	T	.	.	.	.	.	.	.	.	.	.	.	.	.	.	G	.	.	.	.	.	.	.	.	A	.	.	1	
H26	.	.	.	.	.	.	.	.	.	.	.	.	.	.	.	.	.	.	.	.	.	.	.	.	.	.	.	.	.	.	.	.	.	.	.	C	.	.	.	.	.	.	.	1	
H27	.	C	.	.	.	.	A	.	.	.	.	.	.	.	.	.	.	.	.	.	.	.	.	.	.	.	.	.	.	.	.	.	.	.	.	.	.	.	.	.	.	.	.	1	
H28	.	C	.	.	.	.	.	.	.	.	.	.	C	.	.	.	.	.	.	.	.	.	.	.	.	.	.	.	.	.	.	.	.	.	.	.	.	.	.	.	.	.	T	1	
H29	C	C	.	.	.	C	A	.	.	.	.	.	.	.	.	.	.	.	.	.	.	.	.	.	.	.	.	.	.	.	.	.	G	.	.	.	.	.	.	.	A	.	.	1	
H30	C	C	.	.	.	C	A	.	.	.	.	.	.	.	.	.	.	.	.	.	C	.	.	.	.	.	.	.	.	.	.	.	.	.	.	.	.	.	.	.	A	.	.	3	
H31	.	C	.	T	.	.	.	.	.	.	.	.	C	.	.	.	.	.	.	.	.	.	.	.	.	.	.	.	.	.	.	.	.	.	.	.	.	.	.	.	.	.	.	1	
H32	.	.	.	.	.	.	.	.	.	.	.	.	.	.	.	.	.	.	.	G	C	.	.	.	.	.	.	.	.	C	G	.	.	.	.	.	.	.	.	.	.	.	.	1	
H33	.	.	.	.	.	.	.	.	.	.	.	.	.	.	.	.	T	.	.	G	.	.	.	.	.	.	.	.	.	.	.	.	.	.	.	.	.	.	.	.	.	.	.	1	
H34	.	C	.	.	.	.	.	.	.	G	.	.	C	.	.	.	.	.	.	.	.	.	.	.	.	.	.	.	.	.	.	.	.	.	.	.	.	.	.	.	.	.	.	1	
Notes.

n number of observations of each haplotype

Table 2 Genetic diversity of Tachypleus gigas in each locations.

Population	Code	A+T%	n	Nh	h	π	
Bintan	BT	81.597	8	6	0.892	0.006	
Balikpapan	BP	81.473	14	10	0.945	0.009	
Demak	DK	81.568	16	6	0.783	0.004	
Madura	MD	81.412	13	8	0.910	0.006	
Subang	SB	81.548	20	11	0.926	0.005	
Ujung Kulon	UK	81.434	20	12	0.942	0.005	
Total		91		0.935	0,0064	
Notes.

n number of samples

Nh number of haplotype

h haplotype diversity

π nucleotide diversity

At a glance, the obtained haplotype diversity was high, ranging from h = 0.783 to 0.945 with a mean gene diversity per population h = 0.935. Conversely, the nucleotide diversity was relatively low in all locations, ranging from π = 0.004 to 0.009. The overall diversity was similar across populations. DK had the lowest haplotype and nucleotide diversity (h = 0.783, π = 0.004). BP had the highest haplotype and nucleotide diversity (h = 0.945 π = 0.009), followed by UK (h = 0.942, π = 0.005), SB (h =0.926, π = 0.005), MD (h =0.910, π = 0.006), and BT (h = 0.892, π = 0.006) (Table 2).

Population structure

Pairwise FST values ranged from 0 to 0.13 across the populations (Table 3). Generally, the FST value among locations was not significantly different from zero (p > 0.05) with the exception of UK-MD and UK-SB, indicating the restricted gene flow among these populations. Populations with higher pairwise FST values included BT-MD (p > 0.05), BT-SB (p > 0.05), UK-MD (p < 0.05), and UK-SB (p < 0.05). The pairwise FSTvalues of UK-BT, DB-DK, and SB-MD were effectively zero. Our AMOVA results showed that the majority of variation was found within (95.23%) rather than among (4.77%) populations (Table 4).

Table 3 Pairwise FST between populations of Tachypleus gigas in six sampling locations.

 	BT	BP	DK	MD	SB	UK	
BT	–						
BP	0.05	–					
DK	0.08	0.00	–				
MD	0.13	0.00	0.00	–			
SB	0.11	0.01	0.00	0.00	–		
UK	0.00	0.08	0.09	0.10*	0.10*	–	
Notes.

FST value significantly different (p < 0.05)*.

BT Bintan

BP Balikpapan

DK Demak

MD Madura

SB Subang

UK Ujung Kulon

Table 4 The analysis of molecular variation (AMOVA) that conducted based on the haplotype frequencies of Tachypleus gigas.

Source of variation	d.f	Percentage of variation	FST	p-values	
Among populations	5	4.77	0.04	0.006	
Within populations	85	95.23			
Total	90				

Population connectivity

The relationship of the 34 haplotypes was illustrated using a median-joining network (Fig. 2). The haplotype network showed that there were shared haplotypes (H1, H3, H5, H6, H8, H9, and H18) across the geographic sites. H3 was the most common, and was identified in all populations except UK and including 15 individuals. H5 was found in 12 individuals from the BT, BP, DK, SB, and UK populations. However, specific haplotypes were only found in certain locations. The UK population had the highest number of specific haplotypes (seven). Meanwhile, BT had the lowest number of haplotypes (two) (Fig. 3).

Figure 2 Haplotype network of Tachypleus gigas (n = 91) population in six locations around Indonesia, constructed with Median-Joining method.

Figure 3 Distribution of 34 haplotypes of Tachypleus gigas population from six locations in around Indonesia.

We assessed historical demography based on mtDNA AT-rich region haplotype frequencies. There were shared haplotypes in all locations (Fig. 2). Furthermore, the Tajima’s D test values (Table 5) were negative across all populations, with the exception of DK, MD, and SB. They showed no significant p-values, indicating that there was no evidence of selection. Similarly, the Fu’s F s test results (Table 5) were negative (except in DK), with no significant p-values across all six populations. This indicated an excess number of haplotypes, as expected due to a recent population expansion.

Discussion

In this study, there was high haplotype diversity in six coastal horseshoe crab populations in the northern Java Sea, Bintan, and Balikpapan waters of Indonesia. There was also a high number of polymorphic sites (43, with 34 defined haplotypes) in Indonesian coastal horseshoe crab populations. The mean haplotype diversity (h = 0.935) was quite high, while nucleotide diversity (π = 0.006) was low across all populations. Similarly high haplotype diversity values were reported in T. gigas (h = 0.797 ± 0.129 and π = 0.058 ± 0.001; Rozihan & Ismail, 2011) in Malaysia and tri-spined horseshoe crab (T. tridentatus) in Taiwan (h = 0.626 ± 0.075 and π = 0.003 ± 0.005; Yang et al., 2007).

Previous studies reported generally high genetic diversity in coastal horseshoe crab (Rozihan & Ismail, 2011; Aini et al., 2020). Our results showed not only high genetic diversity, but also low nucleotide diversity. The high number of haplotypes indicates that these populations were large enough to maintain a high level of genetic diversity. These small differences are the signature of rapid demographic expansion from a small effective population size (Avise, 2000). Nucleotide diversity is a sensitive index when analyzing population genetic diversity (Nei & Li, 1979), and is influenced by life-history characteristics, environmental heterogeneity, population size (Nei, 1987; Avise, 2000), fishing pressure (Madduppa, Timm & Kochzius, 2018), level of larval transport, and degree exchange with other populations (Timm et al., 2017). The rate of mitochondrial evolution and historical factors play an important role in determining genetic variability patterns (Grant, Spies & Canino, 2006; Xiao et al., 2009; Yamaguchi, Nakajima & Taniguchi, 2010).

Table 5 Results of Tajima’s D and Fu’s FS tests including associated p-values in all locations.

Population	Tajima’s D	Fu’s FS	
Bintan	−0.646ns	−0.608ns	
Balikpapan	0.601ns	0.847ns	
Demak	0.325ns	−2.941ns	
Madura	−0.875ns	−1.532ns	
Subang	0.166ns	−0.891ns	
Ujung Kulon	−0.318ns	−3.865ns	
Notes.

ns not significant

We detected low differentiation across populations (insignificant FST values between 0 and 0.13), with exceptions between populations UK-MD and UK-SB. This result indicated that there was little subdivision across populations. Several studies suggested restricted dispersal abilities for horseshoe crabs regarding short-term tagging. However, some others explained that this crab has a wide dispersal abilities based on long-term studies. Individual distances up to 30 km have been observed in Malaysian crabs (Mohamad et al., 2019), while the movement abilities of tri-spined horseshoe crab did not exceed 150 km (Yang et al., 2007). Similarly, the American horseshoe crab in the Great Bay Estuary (USA) has a maximum mean annual linear distance ranging between 4.5 km and 9.2 km (Schaller, Chabot & Watson, 2010). Studies by Swan (2005) over multiple years found that Limulus moved from 104 to 265 km from their release sites. Ecological observations showed that their hatched larvae swim freely for approximately 6 days and then settle in the bottom of shallow waters around their natal beaches (Shuster Jr, 1982). However, larvae have a strong tendency to concentrate in inshore rather than offshore waters (100–200 km) (Botton & Loveland, 2003), suggesting a limited ability for long-range dispersal between estuaries. Additionally, low FST levels reflect inter-population movement over mutigenerational intervals that short-term tagging studies cannot document. Long-term tagging studies have found that horseshoe crabs can move from >5–500 predominated km 5–30 km (Beekey & Mattei, 2015), and up to 767 km over their long lifetimes (E. Hallerman, 2020, personal communication). Long-term tagging study similar study by Rozihan & Ismail (2011) reported that the crab’s FST value along the west coast of peninsular Malaysia ranges from 0.111–0.557, indicating moderate to high genetic differentiation (Wright, 1978; Hartl & Clark, 1997). Other reports in the area used microsatellite markers to find a FST value between 0.144 and 0.846.

There were only seven shared haplotypes among the 34 total haplotypes observed among all 91 samples. The median-joining network analysis indicated past population expansions with shared haplotypes among localities. Overall, relationship patterns at the mtDNA level showed little geographical structure. The haplotype network revealed recent demographic processes, but the small sample sizes also limited the possibility of observing the intermediate haplotypes inferred to exist in the network. Moreover, results of Tajima’s D and Fu’s F s tests indicated the occurrence of population expansion. Common haplotypes shared between localities also can be explained by the historical biogeography in this Southeast Asian region known as the Sunda Shelves, which includes Java, Sumatera, and Borneo. Historically, Sundaland experienced both dewatering and inundation during the Pleistocene period. Haplotype sharing in this study is attributed to breeding migration and dispersal of pelagic larvae, as well as the sharing of common ancestors (Frankham, 1996). The occurrence of many geographic site-specific haplotypes can be explained by the small sample size and perhaps historical isolation during the Last Glacial Maximum. Many species became isolated in refugia, and genetic differentiation and divergence occurred due to the retreat and dispersal of glacial ice sheets (Hewitt, 2000).

A proactive management approach regarding the Asian coastal horseshoe crab (T. gigas) in Indonesia should consider population genetics. High haplotype diversity that occurs with low nucleotide diversity has been associated with population growth or expansion after a period of low effective population growth (Grant & Bowen, 1998). Our findings indicate that T. gigas in Indonesia have low genetic differentiation but high population connectivity and expansion. Therefore, our results suggest that there is a single stock of Indonesia coastal horseshoe crab. The best conservation strategy would be one that combines both local and regional management. To expand our knowledge base, an advanced population genetic analysis based on male and female horseshoe crabs and the nuclear genome (e.g., microsatellites or SNPs) should be conducted. This should also include expanding the scope of geographic sampling around Indonesia.

Conclusion

High genetic diversity and low levels of differentiation across coastal horseshoe crab (T. gigas) populations in Indonesia indicated a single stock with high connectivity. A locally and regionally based conservation management method is suggested as a precautionary approach to conserving the Indonesian coastal horseshoe crab.

Supplemental Information

Supplemental Information 1 Sequence data of Tachylpleus gigas from all locations

Click here for additional data file.

The authors are grateful to everyone who helped in the field work: local fishermen in all sampling areas, Dr. Qian Tang, Heri Saputro, Agus Alim Hakim, Rani Nuraisah, Siti Mira Rahayu, Ahmad Fauzi Ridwan, and Yunita Multi Cahya Ningrum.

Additional Information and Declarations

Competing Interests

Author Contributions

Field Study Permissions

DNA Deposition

Data Availability

Hawis Madduppa is employed by Oceanogen Environmental Biotechnology Laboklinikum.

Naila Khuril Aini and Hawis Madduppa conceived and designed the experiments, performed the experiments, analyzed the data, prepared figures and/or tables, authored or reviewed drafts of the paper, and approved the final draft.

Yusli Wardiatno conceived and designed the experiments, analyzed the data, authored or reviewed drafts of the paper, and approved the final draft.

Hefni Effendi analyzed the data, authored or reviewed drafts of the paper, and approved the final draft.

Ali Mashar conceived and designed the experiments, prepared figures and/or tables, authored or reviewed drafts of the paper, and approved the final draft.

The following information was supplied relating to field study approvals (i.e., approving body and any reference numbers):

Field experiments were approved by the Research Council of Study Program the IPB University (Letter number: 1426/IT3.F3.2/KP.03.03/2019).

The following information was supplied regarding the deposition of DNA sequences:

The sequences are available at BOLD (Project - HWSHC Genetics of horseshoe crab (Tachypelus gigas) around major habitats in Sundaland): HWSHC.

The following information was supplied regarding data availability:

The raw measurements are available in the Supplemental File.

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
