# Peer review of "High genetic diversity and mixing of coastal horseshoe crabs (Tachypleus gigas) across major habitats in Sundaland, Indonesia"

_PeerJ, doi:10.7717/peerj.11739_

## Round 0.1 · original submission · Major Revisions

I have now heard back from three reviewers, who all note that while the data included in this paper are of importance, much work is needed to bring the manuscript to an acceptable status. All reviewers have provided constructive comments, and I hope these can guide you in improving your work.

·

Basic reporting

The English prose needs revision - I have marked the manuscript to aid in that task.
The paper lacks citations to relevant studies, which I provide in my review and in a manuscript that I will share.
Some of the inferences must be reconsidered along the lines of my comments below.

Experimental design

The study involves a limited sample analyzed at one mitochondrial locus. There may be a possibility to expand the geographic scale of analysis by inclusion of results of an earlier study as described below. The methods are generally laid out in sufficient detail that subsequent workers can build upon this study.

Validity of the findings

While generally sound, some of the inferences must be reconsidered as described below.

Additional comments

Tachylpelus gigas is one of four horseshoe crab species worldwide. Knowledge of its population genetic structure would inform its management and conservation. Aini et al. collected 91 crabs from six sites in Indonesia, sequenced the AT-rich region of its mitochondrial DNA molecule, and conducted population genetic analyses. The results contribute to our knowledge of population genetic and stock structure of the species. Yet, this study is limited in some critical ways. Sample sizes are small. Just one mitochondrial locus is considered, but no nuclear loci, limiting the scope of inference. The geographic scope of coverage is limited, though it many be possible to combine these results with existing information from another study, as I discuss below. In context below, I will mention several issues with interpretation of biology and analytic results, as well as consideration of all relevant literature. The English prose can be strengthened – I have marked the manuscript to guide revision.
Abstract. – The authors write that the genetic structure of horseshoe crabs is unchanged over millions of years. We don’t know that. I think the authors intend to write that its overall morphology is fundamentally unchanged over hundreds of millions of years. In any case, the sentence is not really needed and can be deleted.
The last sentence of the abstract suggests that adult and juvenile horseshoe crabs exhibit little movement among locations. In the paper, they cite a few studies of another horseshoe crab species, but not this species. To really defend the assertion for T. gigas, they’d have to cite results of long-term mark-recapture studies. Populations genetics indexes movement over recent generations, and mitochondrial studies index only female movement. Males being smaller, they tend to move greater distances. My comments here indicate that this sentence must be carefully revised to prove defensible.
Introduction. – At line 55, I think that the word “movement” may be missing.
At line 66, the issue of genetic structure being unchanged for millions of years arises again.
At line 61, Limulus polyphemus is more frequently referred to as the American horseshoe crab.
At line 69 is a sentence on genetics that seems out of place. I suggest it be moved to line 77. The passage fails to mention two more recent studies of relevance, King et al. (2005) and (2015). Further, an existing study of population genetics of T. gigas by Roihan and Ismail (2011) is relevant in this context, but is not mentioned.
Methods. – It seems that different methods of calculating Fst were employed. Using DNA sequences, Fst as calculated by Rozas et al. (2017) would be preferred. Fst from AMOVA analyses is less precise.
Results. – At lines 166-169, negative values of Fst are not biologically meaningful, they are to be interpreted as essentially zero.
At line 169 and other contexts, geneflow Nm should be reported with the unit of individuals per generation.
At line 176, the authors should not write of “dominant”, but rather “common” haplotypes. Dominance is a descriptor relevant to Mendelian traits.
At line 177, the authors should write that shared haplotypes are an indication of gene flow among geographic sites. Writing of evolutionary links suggests an indication of phylogenetic relationships, which were not considered in this study.
In the last paragraph of Results, the authors write of historical demography and seem to regard their results from the DNA sequence mismatch analysis and from the structure of the haplotype network as conflicting. This is not necessarily so; the DNA sequence mismatch speaks to deeper time, to a demographic expansion perhaps coincident with the post-Pleistocene rise of sea level. The haplotype network structure speaks to more recent demographic processes. I must add though, that small sample size limits the possibility of observing the intermediate haplotypes that are inferred to exist on the network.
Both this study and Roihan and Ismail looked at variation at the same mitochondrial region. If the earlier data are archived, then the authors can combine the existing and new data, trim the sequences to the same length, and repeat the analyses to thereby gain a view of horseshoe crab population genetic structure over a much larger area. This would contribute much to the study.
Discussion. – At line 205, the authors compare the population genetic variation in their study with that observed by Roihan and Ismail – are the respective studies of similar scale? Is this a viable comparison?
At line 212, the authors write that observation of a large number of haplotypes in their data indicates a high mutation rate for mitochondrial genes. This is not a suitable explanation. It is rather that the populations studied are sufficiently large to maintain a high level of genetic diversity, i.e., haplotypes are not being lost to random genetic drift.
Throughout the discussion, the authors do not adequately consider the importance of adult dispersal. There are long-term tagging studies that show that the males in particular can move hundreds of kilometers over their long lifetimes. I will include a manuscript in review – which can be cited as a personal communication - to indicate that. This might be cited at lines 233 and 257.
At line 139, the authors argue that inbreeding may be occurring within horseshoe crab populations. With large populations, considerable dispersal, and late maturity, this does not seem likely. High Fis metrics are more likely the result of Wahlund effect.
At line 240, again negative Fst should be taken as Fst being effectively zero.
At lines 250 and 267, small sample size also explains the results of this study.
At line 253, I do not consider the haplotype network as having the classical start profile cited. It is more complex than that, so explanations would have to be more complex and nuanced.
At line 263, Sundaland was not glaciated and deglaciated, rather it was dewatered and then inundated.
The sentence at line 275 suggests that high mitochondrial sequence variation is adaptive. This is oversold. It may be associated with high nuclear diversity, some of which may be adaptive, but the sentence as written is indefensible.
At line 280, the Walls et al. reference concerns L. polyphemus, another horseshoe crab, so the extension of that assertion to T. gigas seems indefensible as written.
The section should end with a call for a study across the range of the species and including nuclear DNA markers such as microsatellites or SNPs. I found such a call at the end of the Conclusions section. That passage should be moved up. Undiscussed material does not belong in the Conclusion.
Conclusions. – At line 288, the two parts of the sentence do not belong together.
References. – I marked some minor departures from journal citation stylistics.
Tables and Figures. – The footnote to Table 1 does not seem to be cited in the label or body of the table.
Literature cited in this review:
King, TL, Eackles MS, Aunins AW, Brockmann HJ, Hallerman E, Brown BB (2015) Conservation genetics of the horseshoe crab (Limulus polyphemus): allelic diversity, zones of genetic discontinuity, and regional differentiation. Pages 65-96 in Carmichael R, Botton ML, Shin P, Cheung SG, eds. Changing Global Perspectives on Biology, Conservation and Management of Horseshoe Crabs. Springer, Berlin
King TL, Eackles MS, Spidle AP, Brockmann HJ (2005) Regional differentiation and sex-biased dispersal among populations of the horseshoe crab, Limulus polyphemus. Trans Amer Fish Soc 134:441-465

Reviewer 2 ·

Basic reporting

The manuscript requires substantial improvement and language editing
Literatures cited can be improved

Experimental design

Following the aims and scope of the journal
Further improvement in the transparency of the method adopted is to be addressed in the revised version

Validity of the findings

Authors need to argue more on the conclusion made in this study. Please refer the attached file

Additional comments

Author could refer the PDF file attached here for comments and suggestions.
Please do refer 'Mitochondrial DNA Signature for Range-Wide Populations of Bicyclus anynana Suggests a Rapid Expansion from Recent Refugia'

Annotated reviews are not available for download in order to protect the identity of reviewers who chose to remain anonymous.

Reviewer 3 ·

Basic reporting

There is room for the improvement of English language use. Many sentences, especially ones in the introduction and discussion, are unclear. Background and literature references are provided sufficiently. The manuscript's structure is generally good, but it is not necessary to have a single paragraph section of "The Implication for Conservation" outside of "Discussion". The manuscript is mainly a report of observation; therefore, I don't think it is necessary to have a hypothesis. It seems the authors emphasizes the genetic connectivity a lot as the major contributor to the low genetic differentiation. However, population genetic analyses with a single monoploid sequence is insufficient to infer gene flow.

Experimental design

no comment

Validity of the findings

The research is original even though the methodology used is common. The research question is not well defined in the introduction, but it doesn't matter as the results are mainly descriptive. The manuscript certainly addresses some questions and discovered that the horseshoe crabs across multiple Indonesia islands are of low genetic variance, which has not been reported before. The investigation is sound enough and the statistics are robust enough. Methods described provide enough information for replication.

Additional comments

The manuscript is important as the first report to show the shallow genetic differentiation of a marine flagship species across Indonesia. The major limitation is that the use of single genetic marker of monoploidy provide limited resolution to explore further into the causes of such shallow genetic differentiation, such as gene flow, genetic drift and other dramatic demographic processes. I would recommend the publication of the manuscript with substantial revision. Unless authors decide to use additional markers and analyses to support that the shallow genetic divergence is mainly contributed by genetic connectivity, I'd suggest to only preserve "gene flow" as a possible scenario in the discussion section.

Annotated reviews are not available for download in order to protect the identity of reviewers who chose to remain anonymous.

---

## Round 0.2 · Major Revisions

I have heard back from all three reviewers, the same as in the first round of reviews. While all three notice some improvement, they all still have some comments that need addressing. Aside from the scientific comments, all reviewers note the English still needs substantial improvement. Please note English editing is not the job of the reviewers or editor, and it is my opinion that some hard work will be needed to bring your work up to standard; hence my decision of major revisions. I hope you can undertake this task, and look forward to see your revised work.

·

Basic reporting

Please see the general report below

Experimental design

Please see the general report below

Validity of the findings

Please see the general report below

Additional comments

Tachylpelus gigas is one of four horseshoe crab species worldwide. Knowledge of its population genetic structure would inform its management and conservation. Aini et al. collected 91 crabs from six sites in Indonesia, sequenced the AT-rich region of its mitochondrial DNA molecule, and conducted population genetic analyses. The results contribute to our knowledge of population genetics and stock structure of the species. This is revised manuscript, and while improved, still needs clean-up of scientific presentation and prose. I have marked the manuscript to guide revision.
Title page. – Shouldn’t the name of IPB University be spelled out?
Abstract. – At line 29, the sequences were amplified using PRIMERS FOR mitochondrial ….
At line 33, the AMOVA SHOWED MOST VARIATION within (95.23%) rather than among…
At line 37, the lack of genetic structure argues for regional, as opposed to local-based conservation. In that same line, this is a study of modest scope and so its conclusions must not be oversold – the authors should delete “conclusively”.
Introduction. – Throughout the review of this paper, there has been a contention between myself and the authors about how much the crabs move. There have been but a few marking studies, with recaptures likely limited over space and time, that the authors repeatedly cite to argue that movement is little. Yet, longer-term studies of other horseshoe crabs and their own genetics results suggest considerable movement. Part of the issue is that genetics marks movement over a multigenerational period. The authors must distinguish between individual movement as measured by marking and inter-generational migration measured with genetic structure. This issue first comes up at line 54 where the authors write that life-history and habitat use “indicate” restricted dispersal capacity. The Sekiguchi paper cited is a marking study, hence, the sentence should be modified to write of individual movement or simply toned down to say that these factors “suggest” limited dispersal capacity.
The Sekiguchi (1988) citation should be moved to line 60 and a repeated sentence at line 64 can be deleted.
Methods. – At line 92, the manufacturer of the GeneAiD extraction kit should be given.
At line 102, UV light WAS USED TO OBSERVE the band which indicates…
At line 103, where is 1st Base located?
The sentence at lines 117-118 explaining Fst is not needed – the reader is expected to know that. Similarly, the passage from lines 126-131 explaining the D and Fs metrics is not needed. A few words of explanation in Results tells any reader unfamiliar with the metrics what positive or negative estimates mean.
The supporting (Excoffier and Lischer 2010) citation should appear at line 128.
Results. – The sample sizes in this study are small, yet some genetic diversity metric are presented to four decimal places. While the software does report it, the sample sizes do not justify an implicit claim of four-place precision. The authors should report their metrics to three places (three significant places if there are leading zeros). I’ve marked some, but not all such instances on the manuscript.

At the end of line 147, the authors should report h and pi for the BP population – those values are missing.
At line 164, the sentence should start as: RELATIONSHIPS AMONG 34 haplotypes were ILLUSTRATED… This analysis did not identify haplotypes as written.
At line 175, the authors should tell the reader the number of haplotypes in the UK population.
At line 177, the historical demography was assess based on the FREQUENCIES OF mtDNA AT-rich region haplotypeS.
Discussion. – At line 208, there is LIITLE, not no subdivision among populations. The sentence as written is not correct.
The sentence at lines 221-223 is not relevant to the passage in which it appears and should be moved or removed.
The sentence at lines 223-226 should end the passage on movement and connectivity. The authors might add that Fst reflects inter-population movement over mutigenerational intervals that short-term tagging studies cannot.
The sentence at line 232 regard AMOVA is not relevant to the passage and can be removed.
The sentence at lines 240-242 on movement of horseshoe crabs is not relevant to its passage.
The sentence at line 249 should read: The occurrence of the many unique haplotypes is explained by the SMALL sample size AND POSSIBLY ALSO the ISOLATION during the last glacial maximum.
At lines 261-262, the authors write that all individuals do not move far along the coastline. They have no data to support this interpretation and their genetic results suggest considerable movement among sites. Best to simply delete this sentence.
At line 269, best to start the paragraph with High genetic diversity. The authors should not oversell their results by claiming that their findings are “conclusive”.
References. – I marked some minor departures from journal citation stylistics. The King et al. citations are not in the right place.
Tables and Figures. – Entries in Table 1 should not claim so much precision.
“ns” does not appear in Table 3, so the footnote defining that can be removed.

Reviewer 2 ·

Basic reporting

Many comments raised on the manuscript content is not well addressed in the revised article. Authors need to focus each points and improve the manuscript substantially.

Experimental design

Additional information needed (Refer the review file)

Validity of the findings

Sound, Novel and timely.

Additional comments

I advice authors to send the manuscript for language editing.
Further comments can be seen in review file.

Annotated reviews are not available for download in order to protect the identity of reviewers who chose to remain anonymous.

Reviewer 3 ·

Basic reporting

There is still room for improvement in English. I have made some suggestions directly on the manuscript.

Experimental design

no comment

Validity of the findings

Still, the authors' efforts is insufficient to suggest gene flow to be the only contributor to the low divergence of the coastal horseshoe crabs across Indonesia. I'd suggest to tone down the contribution of gene flow in both abstract and introduction. At the same time, provide alternative scenario such as lack of time for divergence from stock population during the last glacial maximum.

Additional comments

In general, the manuscript as a report for observation of low genetic diversity among Indonesia horseshoe crab populations is satisfying. However, the implication of gene flow to such population genetic structure needs further investigations. Moreover, the language should be further improved.

Annotated reviews are not available for download in order to protect the identity of reviewers who chose to remain anonymous.

---

## Round 0.3 · Minor Revisions

You have made progress on your work, and two reviewers have again supplied some constructive comments. Please also check any attachments, and I look forward to seeing your new version of your work.

·

Basic reporting

Needs some minor corrections as noted below, and would benefit from polishing of the technical prose as marked on the scanned, marked manuscript.

Experimental design

It's a study of limited scope, but it does open the field.

Validity of the findings

With some minor prose revisions, findings will be on the mark.

Additional comments

Knowledge of the population genetic structure of Tachylpelus gigas would inform its management and conservation. Aini et al. collected 91 crabs from six sites in Indonesia, sequenced the mitochondrial AT-rich region, and conducted population genetic analyses. The results contribute to our knowledge of population genetic and stock structure of the species. Yet, this study is limited because sample sizes are small, one mitochondrial locus but no nuclear loci are screened, and the geographic scope of coverage is limited. This is a revised manuscript and is notably improved. The English prose can be further polished by attention to the marks on the scanned manuscript. A few context-dependent comments appear below.
Methods. – So that others can build on the methods successfully employed, manufacturers of key products should be mentioned at lines 88, 93, and 94.
Results. – At line 148 is an apparent error – it seems to me that haplotype H3 was observed in 15 individuals, not what is written that it was observed in all but 15 individuals.
At line 204, another apparent error – seven, not six haplotypes were shared among locations.
The sentence at line 206 should read that relationship patterns at the mtDNA level showed little geographical structure.
Discussion. – In the Conclusion, the authors should write that locally AND REGIONALLY based conservation management is recommended. They did make this change in the abstract, but apparently omitted it here.
References. – I marked some minor departures from journal citation stylistics.
Hasn’t IUCN revised its Red List assessment of the species since 1996?
Tables and Figures. – Table 1 should include a column noting the number of observations of each haplotype.

Reviewer 3 ·

Basic reporting

I think the manuscript is much clearer then the previous version. I still have a few minor suggestions, especially on the discussion section.

1. The language has great improvement from the last version. However, I noticed that the authors using past tense throughout when reporting methods and results. I'd suggest the authors to use present tense in reporting results in the Abstract, Results and Discussion sections.

2. I did some literature search for my own manuscript suggested by a reviewer to find that "dispersal ability" is a more commonly used term than "dispersal capability" in the field of ecology and evolutionary biology. I'd suggest to replace "dispersal capability" with "dispersal ability" throughout the manuscript.

3. Please provide references for the statement in line 170.

4. Delete "very" in line 181.

5. The authors' suggestion of recent expansion and high connectivity of regional T. gigas is valid based on the results of analyses. To support the genetic analyses, authors may want to find evidences from survey data to confirm the dispersal ability of T. gigas. In the third paragraph of discussion, authors seem to list evidences on the dispersal ability of T. gigas, however the logic flow is weak. I assume the authors trying to suggest that T. gigas may disperse further based on long-term tracking in contrast to the previous suggestions of limited dispersal ability by short-term tracking studies. I'd suggest authors to add a few statement of their comments on the dispersal ability of T. gigas.

Experimental design

No comment

Validity of the findings

No comment

Additional comments

The manuscript is well written. The results and conclusion are relatively robust. After some minor changes, as suggested above, I recommend the publication of this manuscript.

---

## Round 0.4 · accepted · Accept

Thank you for your hard work. I am happy to move this into production and look forward to seeing the published version.